# Bifunctional Ag-Decorated CeO_2_ Nanorods Catalysts for Promoted Photodegradation of Methyl Orange and Photocatalytic Hydrogen Evolution

**DOI:** 10.3390/nano11051104

**Published:** 2021-04-24

**Authors:** Jinwen Liu, Li Zhang, Yifei Sun, Yang Luo

**Affiliations:** 1Third Institute of Oceanography, Ministry of Natural Resources, Xiamen 361005, China; jinwenliu@tio.org.cn; 2School of Environment and Civil Engineering, Dongguan University of Technology, Dongguan 823808, China; zzhangyige@163.com; 3College of Energy, Xiamen University, Xiamen 361005, China; yfsun@xmu.edu.cn

**Keywords:** photocatalytic degradation, photocatalytic hydrogen evolution, catalyst, Ag/CeO_2_, oxygen vacancy

## Abstract

The photodegradation of organic pollutants and photocatalytic hydrogen generation from water by semiconductor catalysts are regarded as the of the most promising strategies to resolve the crisis of global environmental issues. Herein, we successfully designed and prepared a series of silver-decorated CeO_2_(Ag/CeO_2_) photocatalysts with different morphologies by a facile hydrothermal route. The physical properties, charge transfer behavior and photocatalytic performances (degradation and hydrogen evolution) over diverse catalysts with nanocubes, nanoparticles and nanorods shapes were comprehensively studied. It was found that the Ag-decorated CeO_2_ nanorods (Ag/R-CeO_2_) demonstrate the best activity for both photocatalytic methyl orange (MO) degradation and photocatalytic H_2_ production reaction with attractive stability during cycling tests, suggesting its desirable practical potential. The superior performance of Ag/R-CeO_2_ can be ascribed to (1) the facilitated light absorption due to enriched surface oxygen vacancies (OVs) and plasmonic Ag nanoparticles on nanorods, (2) the facilitated photo-excited charge carrier (e^−^-h^+^) separation efficiency on a metal/oxide hybrid structure and (3) the promoted formation of active reaction intermediates on surface-enriched Ag and oxygen vacancies reactive sites on Ag/CeO_2_ nanorods. This study provides a valuable discovery of the utilization of abundant solar energy for diverse catalytic processes.

## 1. Introduction

The rapid development of industry leads to the large consumption of fossil fuels, which has brought inevitable environmental issues. Specifically, the massive emission of CO_2_ and water pollution of organic compounds severely threaten ecosystems and human health [1,2,3]. Therefore, the purification of dye-polluted wastewater and the exploration of clean energy carriers such as H_2_ are two huge challenges for humans to pursue sustainable development [4,5].

As the most representative dye, methyl orange (MO) is not decomposable at ambient conditions [6]. Until now, enormous efforts have been undertaken to develop diverse technologies for the efficient degradation of MO [7]. The pioneering work of the photoreduction of organic pollutant using TiO_2_ photocatalyst can date back to 1979 [8], from which photocatalytic degradation has received extensive studies [9,10]. For decades, the photodegradation of MO using semiconductor photocatalysts has been paid immense attention due to the abundance of solar energy, low cost and easy operability of the reaction system [11]. Additionally, diverse kinds of photocatalysts have been reported to be active for photocatalytic degradation. For example, transition metal oxides including ZnO [12], TiO_2_ [13] and Co_3_O_4_ [14] with the optical bandgap smaller than the wavelength of UV irradiation, fast charge transfer efficiency, and suitable VB/CB positions for corresponding oxidation/reduction reactions were reported as efficient photocatalysts. The photogenerated electron (e^−^) and hole (h^+^) can combine with O_2_ and OH^−^ dissolved in water to formulate **·**O_2_^−^ and **·**OH, respectively [6], and all these oxygen-containing species are highly active for the decomposition of organic pollution molecules. Different from MO degradation, mainly using photo-excited h^+^, the photocatalytic water splitting takes advantage of the photo-excited e^−^ to reduce proton in water and generate H_2_ [15]. It is widely accepted that the quantity and prolonged lifetime of these active photo-generated e^−^ and h^+^ is tightly linked to the enhanced photocatalytic performance [16,17].

CeO_2_, with a low price, robustness, nontoxicity and high redox capability, has obtained tremendous attention in the field of optics [18], electrochemistry [19] and thermocatalysis [2,20]. The surface of CeO_2_ crystal is rich with oxygen vacancies (OVs) due to the emergence of the Ce^3+^/Ce^4+^ redox pair, which enables the efficient manipulation of band structure and light absorption [21,22]. Recent research has illustrated that the surface OVs of CeO_2_ nanocrystals can be tuned by controlling its exposure facets and morphology [23,24]. The crystal facet with weak Ce-O bond strength is favorable for the formation of OVs. This discovery offers a brand-new way to manipulate the optical property of CeO_2_ and further the potential photocatalytic performance.

Besides the manipulation of intrinsic property, some precious metal nanoparticles, for instance Pt and Au, were post-deposited onto CeO_2_ as the co-catalyst to accelerate the separation of photo-excited e^−^-h^+^ pairs and delay recombination through h^+^ trapping by the extra generation of extra hydroxyl species [25,26]. For example, Au@CeO_2_ nanocomposite exhibited higher activity and stability in the photocatalytic degradation of methyl orange and methylene blue, while compared to pristine CeO_2_ [27]. Pt/CeO_x_/TiO_2_ was attested to be a promising catalyst for the photocatalysis in water splitting, and the H_2_ productivity was approximately seven times higher than the benchmark of WO_3_ [28]. However, the noble metals, with a high price and scarcity strongly restricted its large-scale utilization [16]. Thereby, the design of an efficient photocatalyst functionalized by a non-noble metal cocatalyst is highly desirable.

It is reported that Ag nanoparticles hold a big promise in wastewater disposal applications due to their controllable biocompatibility and excellent antibacterial activity [29]. Additionally, Ag can serve as the co-catalyst, substituting expensive Au or Pt, to modify the surface properties of semiconductor photocatalysts. For example, Ag-modified ZnO was proved to be an effective catalyst for the photocatalytic organic pollutant degradation. The conversion of Ag to Ag^+^ by a dissolved oxygen molecule resulted in the formation of large amount of active oxygen groups (HO_2_**·** and **·**OH), benefitting the degradation efficiency [30]. However, the traditional post-deposition method such as impregnation or post-photo-deposition usually added complexity to the fabrication process, which also led to the uneven distribution of Ag particles and poor metal–oxide adhesion [31]. A new design principle to prepare Ag-decorated CeO_2_ with enriched surface OVs is urgently required, which is expected to promote photocatalytic performance for both MO degradation and H_2_ production.

Following this line of thinking, in this paper, we develop a readily available and convenient one-step solvothermal approach to fabricate a series of Ag-modified CeO_2_ photocatalysts with the morphologies of nanocubes, nanoparticles and nanorods, respectively. By applying a set of characterizations and measurements, it was discovered that the Ag-decorated CeO_2_ nanorods possess superior performance and stability for not only photocatalytic degradation, but also hydrogen evolution. The underlying mechanism of improved performance was presented based on the systematic characterization results, which is suggestive and constructive for the future design of novel semiconductor photocatalysts.

## 2. Materials and Methods

### 2.1. Materials

The chemicals utilized in this work were analytical reagent grade and underwent no further purification. Deionized (DI) water was used throughout the study. Ce(NO_3_)_3_·6H_2_O, Ag(NO_3_)_2_, NaOH, NaBH_4_ and ethanol were received from Sinopharm Chemical Reagent Co., Ltd, Shanghai, China.

### 2.2. Photocatalysts Preparation

**Synthesis of CeO_2_ catalyst** CeO_2_ were fabricated via a hydrothermal method [25]. To obtain CeO_2_, Ce(NO_3_)_3_·6H_2_O (3.48 g) was dissolved in 5 mL of DI water (A) and NaOH was dissolved in 35 mL of DI water (B). Then, the solutions A and B were mixed well with magnetic stirring. After 30 min, the as-obtained white solution (A + B) was poured into a Teflon lined stainless steel autoclave with 100 mL volume without stirring, and the hydrothermal treatment was performed for 24 h. Afterwards, the autoclave was cooled to room temperature naturally. The obtained white production was gathered by centrifuging and was alternately washed thoroughly via ethanol and deionized water for several times until the pH = 7. Finally, the composites were kept in an oven at 80 °C for 12 h to get CeO_2_. The amount of NaOH and hydrothermal treatment temperature were changed for different shapes of CeO_2_ (nanorod: NaOH = 9.8 g, temp = 100 °C, nanocube: NaOH = 9.8 g, temp = 180 °C, nanoparticle: NaOH = 0.014 g, temp = 100 °C). The CeO_2_ shapes of nanorods, nanocubes and nanoparticles were marked as R-CeO_2_, C-CeO_2_ and P-CeO_2_, respectively.

**Synthesis of Ag/CeO_2_ catalyst** Ag/CeO_2_ were also fabricated via a hydrothermal method. Similar to the method shown above, after we obtained the solution of A + B, 0.082 g of AgNO_3_ was mixed into the solution. After stirring for another 20 min, NaBH_4_ (0.2 g) was finally put into the mixture. Then, the above solution was hydrothermally treated at 180 °C for 24 h in a Teflon lined stainless steel autoclave (100 mL). The obtained production was gathered by centrifuging and was alternately washed via ethanol and deionized water until the pH = 7. Finally, the composites were put down in an 80 °C oven and kept there for 12 h to obtain Ag/C-CeO_2_. The synthesis Ag/CeO_2_ nanorods was similar to Ag/C-CeO_2_ except that the temperature was changed from 180 °C to 100 °C. The composite was denoted as Ag/R-CeO_2_, while the synthesis process of Ag/CeO_2_ nanoparticles was similar to Ag/R-CeO_2_ except that the amount of NaOH was changed from 9.8 g to 0.014 g for Ag/P-CeO_2_. The sample was denoted as Ag/P-CeO_2_.

The abbreviations of all samples are listed in Table 1.

### 2.3. Sample Characterization

The X-ray diffraction (XRD) patterns were recorded using an X-ray powder diffractometer (Panalytical X’pert Pro Super, Malvern, UK) along with radiation of Cu Kα. For each sample, 20° to 80° of 2θ (Bragg’s angles) were conducted under a 10 °/min rate. Transmission electron microscopy (TEM) images were received via a TECNAI F-20 apparatus. SEM (Scanning electron microscopy) images were observed using ZEISS-SIGMA. A UV–vis Cary5000 spectrometer (VARIAN Company, Palo Alto, CA, USA) using BaSO_4_ reference was used to obtain UV–visible diffuse reflection spectra (UV–vis DRS). An instrument of F-7000 spectrophotometer was conducted to gain photoluminescence (PL) spectra and the excitation wavelength was 325 nm. Photocurrents (PCs) of the resultant catalysts were observed with a CHI660E electrochemical analyzer, attaching a home-made standard three-electrode quartz cell. Sample suspensions dropped directly and coated onto the precleaned ITO (indium tin oxide) glass surface were used as work electrodes. An SCE (saturated calomel electrode) electrode acted as the reference and the Pt wire served as a counter electrode. Na_2_SO_4_ aqueous solution (0.2 M) without other additives served as the electrolyte. The irradiation light was an Hg lamp (500 W, the accumulated intensity: 110 mW/cm^2^). The work electrode (0.95 cm^2^) was exposed to the Na_2_SO_4_ electrolyte. A bias potential of 0.1 V (vs. Ag/AgCl) was applied. UV–Vis Raman System Invia manufactured by Renisha using a laser Raman spectrometer was performed to observe the Raman test. The excitation source was a 532 nm laser, and the scanning wave number range is specified in this paper.

### 2.4. Catalytic Activity Assessment

The photocatalytic degradation performance of pristine CeO_2_ and Ag/CeO_2_ composites were evaluated by degrading MO under a 500 W Hg lamp (NBT Science Co. Ltd. China, λ = 200–1100 nm, the accumulated intensity: 110 mW/cm^2^) illumination, positioned at a distance of 5 cm over the photocatalytic reactor. Experiments were conducted at ambient temperature. Typically, 50 mg of the resultant photocatalyst was decentralized in a 100 mL MO pollutant solution (5 mg/L, pH = 6.5–7.0). In order to achieve the adsorption and desorption equilibrium of the photocatalyst, MO and water, the above mixture was kept at magnetic stirring (the speed of magnetic stirrer is 300 r/min) for 20 min in the chamber under dark ahead of light irradiation. With magnetic stirring, the suspension was exposed to light irradiation. The collection of 4 mL of solution from the reaction suspension every 20 min was centrifuged to elimination the photocatalyst. Then, the concentrations of MO dye in every sample were analyzed via a UV–vis spectrophotometer at 464 nm (MO characteristic absorbance).

Photocatalytic H_2_ evolution reaction tests of the samples were conducted using a reaction cell reactor with the volume of 500 mL. An amount of 0.10 g powder catalyst was decentralized in a mixture solution containing 180 mL of H_2_O, H_2_PtCl_6_ solution and 20 mL of methanol (as a sacrificial reagent). H_2_PtCl_6_ solution was photo-reduced to obtain Pt (0.5 wt%) cocatalyst modified catalyst in the water splitting experiments. Then, the reactor was degassed via a pump. To establish the balance between adsorption and desorption, the mixture was kept for 20 min (300 r/min) under magnetic stirring prior to irradiation. The light source was a 300 W Xe lamp (Beijing China Education Au—Light Co., Ltd., Beijing, China, the accumulated intensity: 450 mW/cm^2^), positioned at a distance of 5 cm over the photocatalytic reactor. For the sake of maintaining the experiment temperature, a water cooling system was used around the photoreactor. A thermal conductivity detector from GC 2060 gas chromatograph served to monitor the amount of H_2_. Additionally, the carrier gas was high-purity N_2_.

## 3. Results and Discussion

First, XRD characterization was employed to explore the crystal structure of various catalysts, and the results are shown in Figure 1. The diffraction peaks at 28.5, 33.1, 47.5, 56.3, 59.1, 69.4, 76.7, and 79.1 degrees emerged on all CeO_2_ samples can be assigned to (111), (200), (220), (311), (222), (400), (331), and (420) CeO_2_ planes, respectively, suggesting the pure cubic fluorite structure (JCPDS 81-792) of all the CeO_2_ [32,33]. Interestingly, the sharper diffraction peak could be seen on CeO_2_ cubes, which could be ascribed to its larger crystal size [25]. In comparison, much weakened and broader diffraction peak are observed on R-CeO_2_, suggesting the low crystallinity and partially disordered modality of CeO_2_ with a nanorod shape.

After introducing Ag, the diffraction peak of Ag (111) at 38.6° (the purple rectangle area in Figure 1) emerged on Ag/R-CeO_2_ and Ag/C-CeO_2_, evidencing the successful decoration of Ag. From the perspective of apparent optical property, the CeO_2_ nanocubes showed a gray–white color, while the nanoparticles and nanorods samples were yellow and bright yellow, respectively. Nevertheless, after introducing Ag, those samples’ colors significantly changed, indicating the alternation of optical property. Specifically, Ag/C-CeO_2_ became light purple and Ag/R-CeO_2_ turned to dark purple (inset of Figure 1a).

SEM was then utilized to characterize the detailed morphology of as-prepared CeO_2_ and Ag/CeO_2_ nanocomposites. As can be seen in Figure 2a, the C-CeO_2_ displayed a quasi-cubic shape with a side length of approximately 20–100 nm. Meanwhile, the diameter of the round-shape CeO_2_ particles was ~50 nm and the length of the CeO_2_ nanorods was between 20–100 nm. The SEM images in Figure 2d–f shows that the CeO_2_ with different shapes maintained their original morphology after the introduction of Ag nanoparticles. Moreover, the detailed morphology and lattice structure of Ag/R-CeO_2_ were characterized by TEM, TEM-EDX and high-resolution TEM (HR-TEM). The Ag/R-CeO_2_ with the rod-shape obtained the uniform diameter of around 20 nm (Figure 3a). The HR-TEM images (Figure 3b) demonstrated a lattice spacing of 0.19 nm, which is assignable to the (110) plane of fluorite structure CeO_2_ [34], while the Ag nanoparticles with a small diameter of around 3 nm could also be detected. The interplanar distance of 0.24 nm is consistent with the Ag (111) plane [35]. Moreover, the interface between CeO_2_ and Ag could be observed clearly. The TEM-EDX images shown in Figure 3c demonstrate the even distribution of all elements, including O, Ce and Ag, which are in line with the above XRD analysis that the Ag nanoparticle is successfully decorated onto the CeO_2_ nanorod.

The UV–Vis DRS spectra of C-CeO_2_, Ag/C-CeO_2_, P-CeO_2_, Ag/P-CeO_2_, R-CeO_2_ and Ag/R-CeO_2_ samples are shown in Figure 4a, which confirms the optical absorption property of the samples with different morphologies. The absorption spectrum of R-CeO_2_ significantly shifts to a higher wavelength, as compared to those of P-CeO_2_ and C-CeO_2_, suggesting that the R-CeO_2_ exhibits a much stronger absorbance in the low frequency region. The Tauc equation was applied to determine the bandgap energy (Eg) of different CeO_2_ [36]:(*αhv*)*^n^* = A(*hv* − Eg)
where *α*, *n*, A and Eg correspond to the absorption coefficient, the light frequency, a constant and optical band gap, respectively. The *n* of CeO_2_ is equal to 2 [37]. The Eg of the CeO_2_ samples can be computed from the (*αhv*)*^n^* versus (*hv*) plot depiction (shown in Figure 4c). The bandgap values are reported in Table 2. The R-CeO_2_ owns the smallest bandgap energy of 3.04 eV, which is much smaller than that of C-CeO_2_ and P-CeO_2_. Before, it was reported that the Eg of CeO_2_ could be manipulated by changing the morphologies and sizes of the crystal [38], and the light absorption ability is closely related to the optical bandgap of the material. Hence, it is reasonable to speculate that R-CeO_2_, with the smallest bandgap, should obtain the highest light absorption capability.

After the decoration of Ag, the greatly boosted visible light (vis-light) absorption (~600 nm) could be observed on Ag-R-CeO_2_ samples. The phenomenon could be explained by the surface plasmon resonance (SPR) effect of Ag, which was also found on other reported Ag modified catalysts [39,40]. On the surface decorated with SPR excitation of Ag metal, light can be captured and confined nearby Ag, and the Ag nanoparticles’ resonance can play the role in improving the absorption of light for semiconductors [35,39,41], and further enhancing the photocatalytic activity.

Raman spectroscopy with a 532 nm excitation laser was further employed to disclose the vibration information of metal–oxygen bond, as demonstrated in Figure 4b. The Raman spectra of all CeO_2_ possess the characteristic peaks at 462 cm^−1^ which can be imputed to the F_2g_ symmetric stretching vibrations pattern of fluorite CeO_2_ structure [41]. The peak at 530–600 cm^−1^ could be assigned to the band of defect-induced (D), which is directly linked to the presence of defects or OVs in CeO_2_ [42]. The relative intensity ratio of *I_D_/I_F2g_* can be applied to determine the relative concentration of OVs in different samples. Apparently, this ratio for R-CeO_2_ (*I_D_*/*I_F2g_* = 0.596) is much higher than those of C-CeO_2_ (*I_D_*/*I_F2g_* = 0.056) and P-CeO_2_ (*I_D_*/*I_F2g_* = 0.295) samples, suggesting that R-CeO_2_ possesses the largest amount of OVs (Figure 4d). After the loading of Ag, the relative intensity ratio of *I_D_*/*I_F2g_* Ag/R-CeO_2_ sample drastically increases to 1.014, which is around two times larger than that of pure R-CeO_2_, indicating the surging content of OVs. This is partially due to the additive of the highly reducible agent of NaBH_4_ during fabrication. Additionally, this phenomenon was in line with the results reported in the previous literature [34,43,44] that the loaded noble metal (such as Ru, Ag, Pt) could facilitate the generation of OVs [34,35,45]. The surface decoration of Ag nanoparticles is able to activate the surface lattice oxygen to create more OVs [34].

The photocatalytic property of the CeO_2_ and Ag/CeO_2_ with various morphologies were utilized for the photocatalytic MO dye degradation, and the results are shown in Figure 5a. The blank trial of the decolorization of MO dye verifies that only 15% and 12% of MO can be naturally degraded without catalysts (irradiation only) or without light irradiation (with Ag/R-CeO_2_ only), respectively. This result could be explained by the self-photosensitized reaction of MO dye and absorption process of Ag/R-CeO_2_ [31,46,47]. When both light irradiation and catalysts were applied, the degradation efficiency of MO was remarkably promoted, implying the synergistic effect between catalyst and light excitation. All the CeO_2_ nanocomposites exhibit moderate photocatalytic activity, and the R-CeO_2_ decreased the concentration of MO by ~60% after 160 min treatment. This performance was much better than that of CeO_2_ nanoparticles (~20%) and CeO_2_ nanocubes (~50%). After the introduction of Ag, the degradation rate of MO dye was further improved on all samples. More than 80% of MO was decomposed for the sample Ag/R-CeO_2_ after light irradiation for 160 min, whereas about 41% of MO can be degraded over Ag/C-CeO_2_ and 59% of MO be degraded over Ag/P-CeO_2_. The kinetic linear modeling results of CeO_2_ and Ag/CeO_2_ are shown in Figure 5b; the degradation process followed a first order kinetic [48], and the Ag/R-CeO_2_ displays the largest kinetic constant of 0.00969 min^−1^, suggesting its highest conversion rate. To further confirm the degradation performance on Ag/R-CeO_2_, the time-dependent UV–vis absorption spectra of MO concentration on Ag/R-CeO_2_ are shown in Figure 5c. The wavelength of about 420 nm is confirmed for the absorbance of MO. Obviously, the intensity of MO absorbance declined rapidly as the treatment time increased, indicating the consumption of MO. Additionally, based on the result of UV–vis absorbance spectra, no other absorbance band could be detected, revealing that the conjugated structure of MO is destroyed molecules and no other intermediate product is formed [46,49,50].

The stability of the photocatalytic property for the MO degradation over Ag/R-CeO_2_ was further investigated via cycling experiments. Five continuous tests were conducted without renewing the photocatalyst. The procedure of each independent measurement was identical except for refilling the MO solution for each run. After each measurement, the Ag/R-CeO_2_ catalyst was re-collected by centrifugation, rinsed and dried for next cycle. It can be seen from Figure 6 that only a slight drop of degradation performance of MO is observed after five cycles, suggesting the desirable stability of the catalyst. This may be due to the loss of the catalyst during each round of catalyst collection, which is universal in the previous literature [51].

The measurement of the photocatalytic activity of hydrogen evolution on different CeO_2_ and Ag/CeO_2_ catalysts were also performed, and the results are given in Figure 7. R-CeO_2_ shows the highest performance under the simulated solar irradiation. The average hydrogen evolution rates are 157, 109 and 56 μmol h^−1^ g^−1^ for R-CeO_2_, P-CeO_2_ and C-CeO_2_, respectively (shown in Figure 7b). This order is consistent with that of the OV content on different samples. In addition, after the introduction of Ag, the hydrogen evolution rate was significantly improved, and the rate of H_2_ production for Ag/R-CeO_2_ rose from 157 to 316 μmol h^−1^ g^−1^ for Ag/R-CeO_2_. These results indicate that besides OVs, the Ag also exhibits another prominent influence on such an enhancement of performance.

Usually, photo-excited charge separation efficiency is the key effect in determining the photocatalytic degradation performance and hydrogen evolution of a semiconductor photocatalyst. Therefore, to figure out the mechanism of promoted activity on Ag/R-CeO_2_, the separation efficiency of photo-induced charge carriers was performed by the PL analysis at room temperature with an excitation wavelength of 325 nm. The PL spectra for all as-prepared samples are displayed in Figure 8a. Obviously, a shoulder emission peak of around 400 nm can be detected with the excitation wavelength at 325 nm, which stems from the intrinsic luminescence of CeO_2_. The emission intensity of Ag/CeO_2_ is much inferior to that of CeO_2_, suggesting that the recombination of carriers is suppressed after the loading of Ag. Our experimental observations are in agreement with a previous report that the deposition of Ag metal nanoparticles can accelerate the separation and transportation of photogenerated carriers, further resulting in the quenching effect of PL intensity [19,52]. Additionally, compared to Ag/P-CeO_2_ and Ag/C-CeO_2_, the sample Ag/R-CeO_2_ has lowest emission intensity, meaning that the OVs can also help prevent the recombination of photo-excited charge carrier. The transient photocurrent (PC) of the samples was also exploited to investigate the excitation and transfer of photo-induced e^−^ and h^+^. As shown in Figure 8b, upon irradiation, a quick photocurrent response was clearly given on all the samples and the transient PC density of Ag/CeO_2_ is much higher than the other two CeO_2_ during the continuous several light on–off cycles. In addition, Ag/R-CeO_2_ possesses the highest PC density in all Ag/CeO_2_ samples, meaning the larger amount of photoinduced electrons on the surface could be rapidly transferred.

Based on the analysis above, the enhanced photocatalytic degradation/hydrogen evolution performance of the Ag/R-CeO_2_ nanorods’ photocatalyst can be understood and explained by the possible mechanism proposed in Figure 9. Upon the photocatalyst being irradiated by the Hg lamp, CeO_2_ was activated to produce the e^−^ and h^+^. First of all, the Ag/R-CeO_2_ with rich OVs and plasmonic Ag nanoparticles narrowed the optical bandgap and benefits the light absorption. Secondly, both OVs and Ag facilitated the generation of a larger amount of photo-excited charge carrier and suppressed their recombination simultaneously. Then, these alive e^−^ and h^+^ with prolonged lifetime combined with water-dissolved O_2_ and OH^−^ to create highly oxidative **·**O_2_^−^ and **·**OH to accelerate the decomposition of MO.

In addition, the surface OVs may play another important role in the absorption of dissolved oxygen molecule, which can furnish more effective surface reaction sites for photocatalytic degradation, while for the photocatalytic water splitting process, the accumulated photoelectrons quickly transferred to surface Ag and Pt nanoparticles to react with protons for hydrogen production. Therefore, the photocatalytic hydrogen evolution performance can be significantly enhanced.

## 4. Conclusions

In this work, we successfully synthesized a series of Ag/CeO_2_ nanocubes, nanoparticles and nanorods’ photocatalysts via a facile one-step hydrothermal process. The photocatalytic degradation of MO and photocatalytic hydrogen evolution reaction of the catalysts demonstrate strong morphology dependence and the Ag/CeO_2_ nanorods possess the highest photocatalytic activity for both reactions due to the following reasons: (1) the catalyst with a different shape shows a distinctive surface OV concentration and the CeO_2_ nanorods demonstrate the largest amount of OV concentration. The OVs and surface decoration of Ag with the SPR effect collectively leads to the narrowing of the optical band gap, the increase in light absorption, the prolonged lifetime of the charge carrier and the improvement of photocatalysis performance. (2) The decoration of Ag nanoparticles on CeO_2_ nanorods can also prevent the recombination of e^−^-h^+^ pairs by efficient electron transfer from CeO_2_ to Ag, leading to the charge separation efficiency and enhanced photocatalytic hydrogen evolution performance. (3) Moreover, both Ag and OVs can serve as a reactive site to facilitate the adsorption of O_2_ and OH^−^ to form highly active **·**O_2_^−^ and **·**OH for photocatalytic degradation. This study is believed to shed light on the exploration of well-designed photocatalysts toward diverse photocatalytic processes.

## Figures and Tables

**Figure 1 nanomaterials-11-01104-f001:**
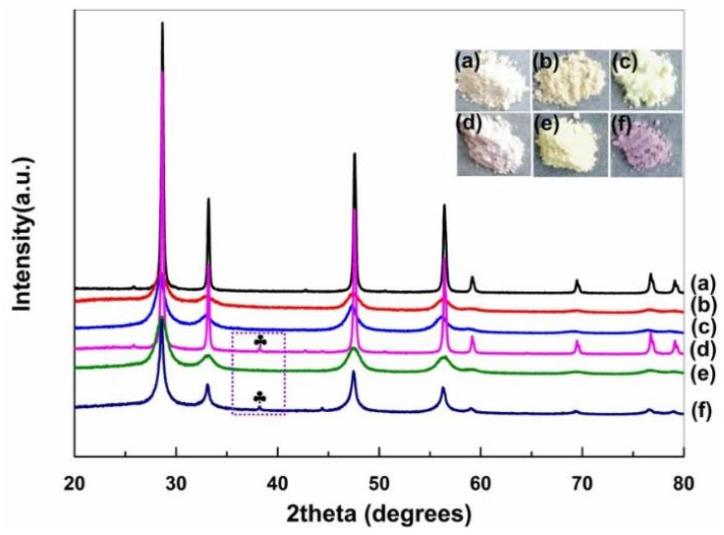
Photographs and XRD patterns of C-CeO_2_ (**a**), (**b**) P-CeO_2_, R-CeO_2_ (**c**), (**d**) Ag/C-CeO_2_, Ag/P-CeO_2_ (**e**), and (**f**) Ag/R-CeO_2_.

**Figure 2 nanomaterials-11-01104-f002:**
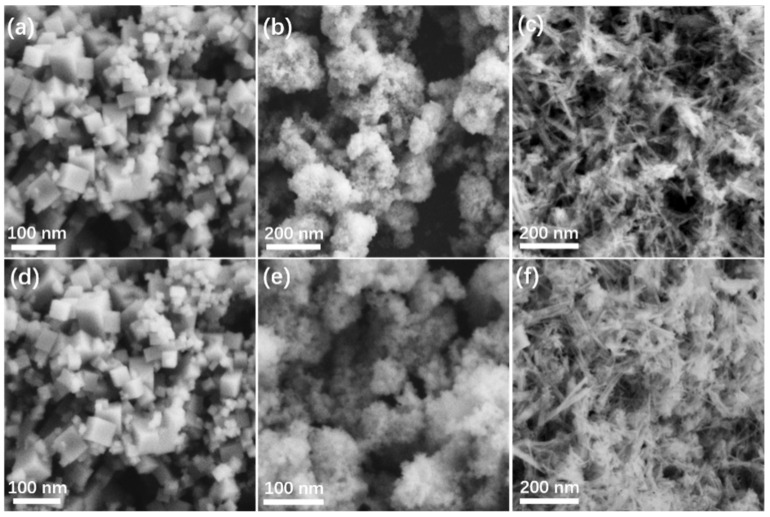
SEM images of various samples (**a**–**f**) corresponding to C-CeO_2_, P-CeO_2_, R-CeO_2_, Ag/C-CeO_2_, Ag/P-CeO_2_, and Ag/R-CeO_2_, respectively.

**Figure 3 nanomaterials-11-01104-f003:**
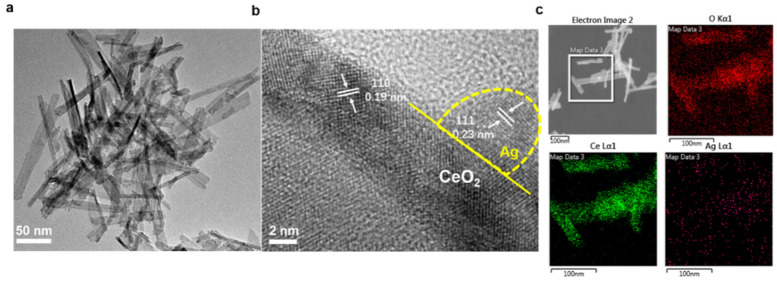
TEM (**a**), HRTEM (**b**) and TEM-EDX (**c**) images of sample Ag/R-CeO_2_.

**Figure 4 nanomaterials-11-01104-f004:**
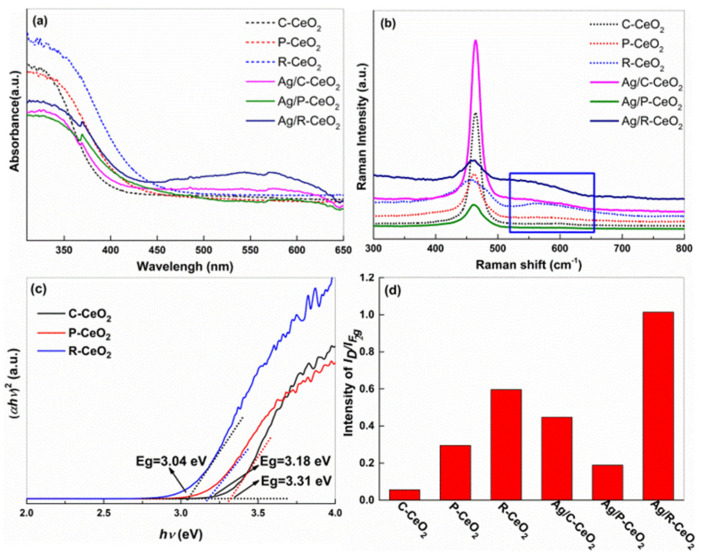
(**a**) UV–vis DRS, (**b**) Raman spectra, (**c**) plot of (*αhv*)^2^ versus *hv* and (**d**) the intensity of *I_D_*/*I_F2g_* in Raman spectra of different CeO_2_ and Ag/CeO_2_.

**Figure 5 nanomaterials-11-01104-f005:**
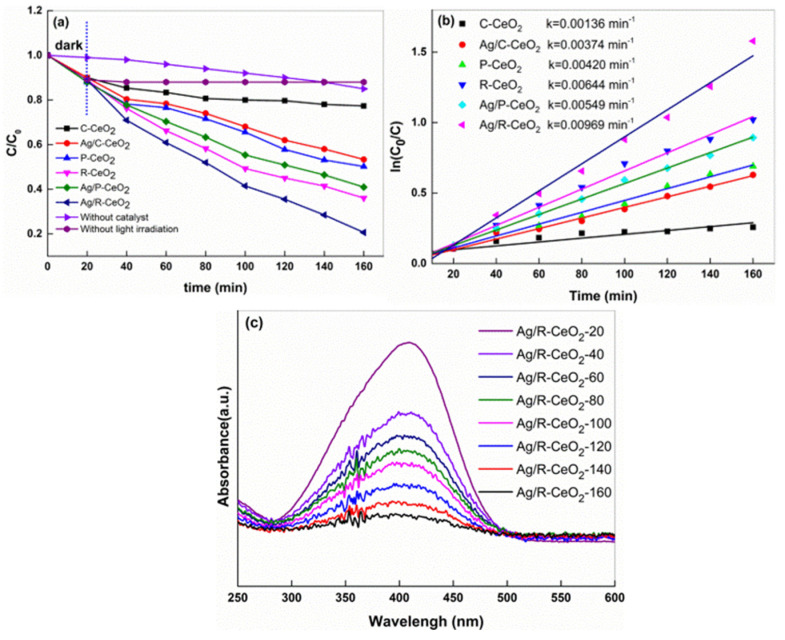
(**a**) The activities over different shapes of CeO_2_ and Ag/CeO_2_ on MO dye, (**b**) kinetic linear modeling and (**c**) UV–visible spectral changes of MO in aqueous at different irradiation intervals for Ag/R-CeO_2_ under light illumination.

**Figure 6 nanomaterials-11-01104-f006:**
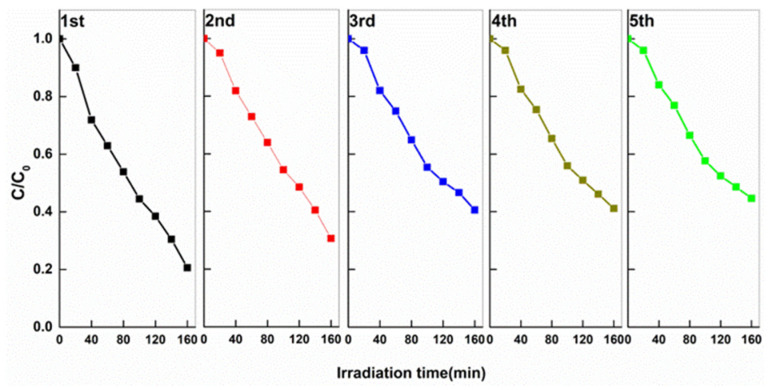
Successive test run for the degradation of MO in aqueous on Ag/R-CeO_2_.

**Figure 7 nanomaterials-11-01104-f007:**
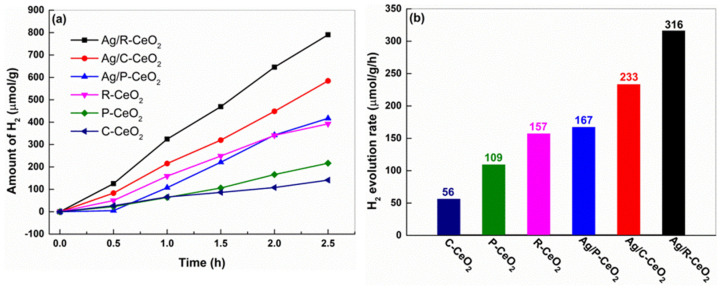
(**a**) The time dependent photocatalytic hydrogen evolution performance plot. (**b**) summary of photocatalytic H_2_ evolution rates over the different resultant samples.

**Figure 8 nanomaterials-11-01104-f008:**
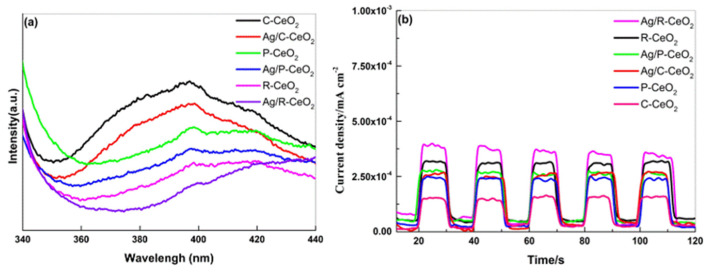
PL emission spectra (**a**) and the photocurrent responses (**b**) of all as-prepared samples.

**Figure 9 nanomaterials-11-01104-f009:**
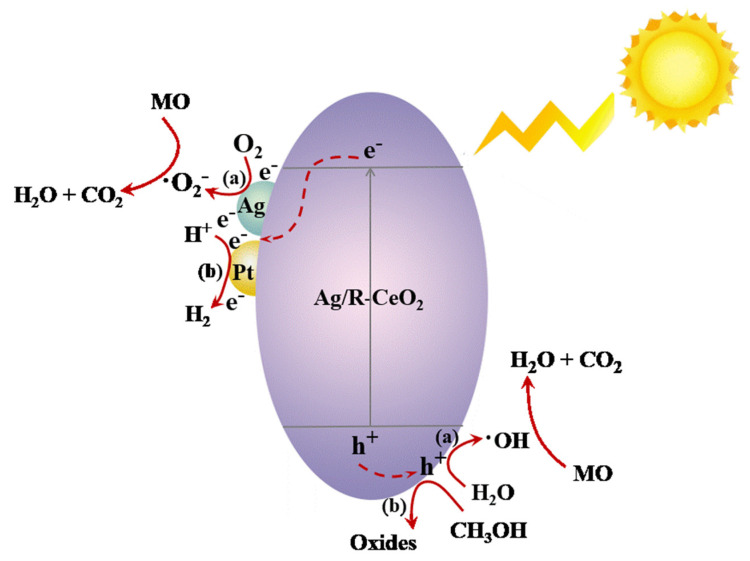
Schematic of the photocatalytic degradation action of MO and photocatalytic water splitting on the Ag/R-CeO_2_ photocatalyst.

**Table 1 nanomaterials-11-01104-t001:** Samples.

Photocatalyst	Abbreviation
CeO_2_ nanocubes	C-CeO_2_
CeO_2_ nanoparticles	P-CeO_2_
CeO_2_ nanorods	R-CeO_2_
Ag decorated C-CeO_2_	Ag/C-CeO_2_
Ag decorated P-CeO_2_	Ag/P-CeO_2_
Ag decorated R-CeO_2_	Ag/R-CeO_2_

**Table 2 nanomaterials-11-01104-t002:** Bandgap values.

Photocatalyst	Bandgap Energy (eV)
C-CeO_2_	3.31
P-CeO_2_	3.18
R-CeO_2_	3.04

## Data Availability

Not applicable.

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
