# Peer review of "Bifunctional Ag-Decorated CeO2 Nanorods Catalysts for Promoted Photodegradation of Methyl Orange and Photocatalytic Hydrogen Evolution"

_nanomaterials, 2021, doi:10.3390/nano11051104_

Round 1

Reviewer 1 Report

The manuscript is interesting, I suggest moderate revisions:

1)English grammar should be double checked;

2)the authors should report a reference for CeO2 nanomaterial production (if it is from literature) or, otherwise, they should justify the mass ratio of ragents, temperature and operative conditions of the syntheses process;

3)the authors should specify: magnetic stirring intensity during photocatalytic experiments, as well lamp source power, wavelength, distance of lamp source from liquid surface of pollutant solution, initial pH;

4)Why did the authora dopted a mass ratio between catalyst and pollutant of 100?

5)the authors should report at least a simplified kinetic data modelling of the obtained results, to estimate kinetic constants that are useful for readers to make quantitative comparisons with literature data (refer to 

Journal of Materials Science: Materials in Electronics, 2021, 32(4), pp. 5082–5093, 
Chemical Engineering and Technology, 2019, 42(2), pp. 308–315 and  Journal of Electronic Materials, 2018, 47(4), pp. 2215–2224 as references, also to enlarge results discussion and introduction part); 6)conclusion section should better underline the advancement of knowledge reached by the findings of this study.

Reviewer 2 Report

This study “Bifunctional Ag decorated CeO2 nanorods catalysts for promoted photodegradation of methyl orange and photocatalytic hydrogen evolution” focuses on designed and prepared a series of silver decorated CeO2(Ag/CeO2) photocatalysts with different morphologies. Here are my comments and questions:

  • You need to improve the introduction. You need literature review on this topic. There are so many recent papers on oriental lacquers using atomic force microscopy. I would recommend authors to include more recent studies in this review. This can help to enhance the innovation meaning and more significance in the field. Also, the authors presented many previous studies, but failed to provide their own analysis based from these studies.
  • Your manuscript is somehow chaotic. It is hard to read. Especially section Results and Discussion, I have started to skip some part because it was hard to read. You have a lot of information there, but you need to improve your story.
  • The quality of Figure 9 needs to be improved.
  • Page 8, what do you mean " … Eg of metal oxide could be manipulated by the morphologies and sizes of the crystal... "?
  • You should check more literature as most of your conclusions have been already addressed by other researchers.
  • Technical terms are misused through the manuscript and the writing needs a minor revision.
  • Page 7, the sentence is unclear: “revealing that the conjugated structure is completely 288 destroyed about MO molecules and no other intermediate product is formed...”; What do authors mean by “no other intermediate product”? Please describe completely the mechanism.
  • The manuscript needs to abbreviations table.
  • Synthesis of CeO2 catalyst need to references
  • Based on its current shape, I recommend a major revision.

Round 2

Reviewer 2 Report

Title: Bifunctional Ag decorated CeO2 nanorods catalysts for promoted photodegradation of methyl orange and photocatalytic hydrogen evolution

In this revised manuscript, the Authors have made corrections according to referee comments. In my opinion, the manuscript in current form could be considered for acceptance.